# Comparison of Two Paradigms Based on Stimulation with Images in a Spelling Brain–Computer Interface

**DOI:** 10.3390/s23031304

**Published:** 2023-01-23

**Authors:** Ricardo Ron-Angevin, Álvaro Fernández-Rodríguez, Clara Dupont, Jeanne Maigrot, Juliette Meunier, Hugo Tavard, Véronique Lespinet-Najib, Jean-Marc André

**Affiliations:** 1Departamento de Tecnología Electrónica, Universidad de Málaga, 29071 Malaga, Spain; 2ENSC, Bordeaux INP, 33400 Bordeaux, France; 3Laboratoire IMS, CNRS UMR 5218, Cognitive Team, Bordeaux INP-ENSC, 33400 Talence, France

**Keywords:** brain–computer interface (BCI), event-related potential (ERP), row-column paradigm (RCP), stimulus, face, picture

## Abstract

A P300-based speller can be used to control a home automation system via brain activity. Evaluation of the visual stimuli used in a P300-based speller is a common topic in the field of brain–computer interfaces (BCIs). The aim of the present work is to compare, using the usability approach, two types of stimuli that have provided high performance in previous studies. Twelve participants controlled a BCI under two conditions, which varied in terms of the type of stimulus employed: a red famous face surrounded by a white rectangle (RFW) and a range of neutral pictures (NPs). The usability approach included variables related to effectiveness (accuracy and information transfer rate), efficiency (stress and fatigue), and satisfaction (pleasantness and System Usability Scale and Affect Grid questionnaires). The results indicated that there were no significant differences in effectiveness, but the system that used NPs was reported as significantly more pleasant. Hence, since satisfaction variables should also be considered in systems that potential users are likely to employ regularly, the use of different NPs may be a more suitable option than the use of a single RFW for the development of a home automation system based on a visual P300-based speller.

## 1. Introduction

The brain–computer interface (BCI) is a technology that uses brain activity to create a gateway between the brain and the external environment. This technology may be especially important for users with severe motor impairments such as amyotrophic lateral sclerosis (ALS) [1]. The most widely studied devices employing BCI technology are spellers, which enable communication through the selection of characters from a display. The most frequent control signal used by BCI spellers is brain activity recorded via electroencephalography (EEG) [2], as the main advantages of this method are that it captures brain signals in a non-invasive way, with a suitable temporal resolution [3]. Previous BCI spellers based on EEG have used specific signals, such as event-related potentials (ERPs), steady-state visual evoked potentials (SSVEPs), or sensorimotor rhythms (SMRs) (e.g., Zhang et al. [4], Nakanishi et al. [5], and Cao et al. [6], respectively). The EEG signals most commonly used by BCI spellers are the ERPs, and the present work therefore focuses on this signal. Of the ERPs, such as the P100, N200, or N400 components, the most important is the P300, and for this reason, ERP-based spellers are typically referred to as P300 spellers. P300 is a positive potential with a latency of roughly 300 ms, and it can be elicited in response to task-relevant stimuli [7].

Farwell and Donchin [8] designed the precursor to the P300 speller and were the first to develop this technology. Their speller was composed of a 6 × 6 matrix containing 36 characters (26 letters and 10 digits). In this paradigm, rows and columns are flashed (i.e., highlighted from grey to white) one by one, and this paradigm is therefore called the row-column paradigm (RCP). To select a character, the user pays attention to the flashing of a specific target character, as this acts as the task-relevant stimulus that elicits the P300 potential. Once the P300 has elicited a specific row and column, the BCI is able to determine the user’s target character.

Improving the usability of the P300 speller is an important goal of the BCI community (see work by Pasqualotto et al. [9] and Guy et al. [10], for example). According to the Organization for Standardization, the definition of usability is composed of three dimensions: effectiveness, efficiency, and satisfaction [11]. Effectiveness represents the completeness and accuracy with which users can complete their objectives. To measure this parameter, error rates or the quality of the solution may be used. Efficiency relates to the resources employed to achieve goals, and indicators of efficiency include the task completion time and task learning time. The last dimension of usability is satisfaction, which reflects the user’s attitude toward and comfort with the system. An attitude rating scale can be used to measure satisfaction, such as the System Usability Scale (SUS) [12]. These three measures should be studied separately, as they are independent aspects of usability [13].

In order to improve the performance of the P300 speller, several visual aspects associated with the presented stimuli have been studied. These visual factors have included the color (e.g., Ryan et al. [14]), the size (e.g., Kellicut-Jones & Sellers [15]), and the use of alternative stimuli, such as faces (e.g., Kaufmann et al. [16]). Kaufmann et al. [16] showed that the use of famous faces as stimuli that appear above the letters can improve performance compared to the use of classical grey-to-white flashing, since this type of stimulus can induce the apparition of several ERPs, notably N170, P300, and N400. Next, Q. Li et al. [17] showed that the use of famous faces colored in green as stimuli improved the performance versus naturally colored faces. It therefore appears that the use of artificially colored faces could provide performance improvements for the P300 speller. Following this, S. Li et al. [18] evaluated the use of red, green, and blue faces, and showed that the use of red faces as stimuli led to superior performance. Finally, Zhang et al. [4] combined the use of a famous face colored in red with a surrounding square of a specific color (white, blue, or red, depending on the experimental conditions). They showed that red faces surrounded by a white rectangle gave the best performance and are thus the best current visual stimulus for use in a P300 speller.

In addition to faces, alternative stimuli have been studied in regard to the control of a P300 speller. Fernández-Rodríguez et al. [19] investigated the emotional properties of pictures and their effects on BCI performance and found that the presentation of pictures above letters, regardless of their emotional content, resulted in better performance compared to the classic use of gray-to-white flashing letters. Unlike the pattern with familiar faces devised by Zhang et al. [4], different images were used for each letter in the paradigm presented by Fernández-Rodríguez et al. [19]. This feature could be useful in the design of P300 spellers, as pictograms could be used to improve and facilitate communication [20].

The aim of the present study is to compare two different paradigms that have already been proven to improve the performance of a P300 speller versus the use of standard gray-to-white flashing letters: (i) one inspired by Zhang et al. [4], in which a single red-colored famous face is displayed surrounded by a white rectangle, and (ii) one based on the scheme of Fernández-Rodríguez et al. [19], in which various neutral pictures are displayed. A preliminary study by Ron-Angevin et al. [21] (N = 4) found that both paradigms may be similar in terms of effectiveness. It was therefore considered appropriate to carry out a more comprehensive assessment of this comparison. In particular, the sample size was increased, and a more complete approach to usability was used, i.e., with measures of effectiveness, efficiency, and satisfaction. The main benefit of the usability approach is that it allows a more comprehensive assessment to be carried out by considering certain dimensions that may be of particular relevance for practical real-life scenarios in which users actually would want to use these interfaces. Moreover, our study permitted us to evaluate the issues of usability and user-friendliness, as suggested by Zhang et al. [4].

## 2. Methods

### 2.1. Participants

The study initially involved 14 healthy French-speaking students from the École Nationale Supérieure de Cognitique de Bordeaux—where the experiment was carried out—although two of these were discarded due to unusable EEG calibration data. The remaining sample contained 12 participants (aged 21.2 ± 1.5, four male and eight female, referred to here as S1 to S12). All of them had normal or corrected-to-normal vision, and they provided written consent. The study was approved by the Ethics Committee of the University of Malaga and met the ethical standards of the Helsinki Declaration. All participants were of legal age. According to self-reports, none of the participants had any history of neurological or psychiatric illness or were taking any medication regularly that could affect the results of the experiment. Before the experiment began, all subjects were informed of the experimental protocol and were able to stop it at any time.

### 2.2. Data Acquisition and Signal Processing

The EEG was recorded via eight electrodes: Fz, Cz, Pz, Oz, P3, P4, PO7, and PO8. All channels were referenced to the right earlobe, using FPz as ground. The signal was modulated by a 16-channel g.USBamp amplifier, which gave a bandwidth from 0.5 to 100 Hz with a sensitivity of 500 µV. A notch filter at 50 Hz was applied, and the EEG was digitized with a sampling frequency of 256 Hz. All aspects of EEG data collection and processing were controlled by BCI2000 software [22]. A stepwise linear discriminant analysis (SWLDA) of the EEG data was performed to obtain the weights for the P300 classifier, to calculate the accuracy, and to enable online spelling.

### 2.3. Spelling Paradigms

The software used to design the layout of the spelling paradigms was the UMA-BCI speller [23], which serves as a user-friendly frontend for BCI2000. Two paradigms were considered in the present study: (i) one involving red faces surrounded by a white rectangle (RFW) and (ii) one with neutral pictures (NP). Each paradigm was evaluated using a 5 × 5 matrix (showing the letters A–Y) using the RCP. The choice of this grid size was made to reduce the number of flashes and to increase the writing speed. As a result, the letter Z did not form part of this study, and none of the written words contained this letter. The background color was black, and the letters on which the stimulus appeared (i.e., faces or pictures, depending on the paradigm) were white. For both paradigms, a stimulus onset asynchrony (SOA) of 281.25 ms and an inter-stimulus interval (ISI) of 93.75 ms were used, meaning that each stimulus was presented for 187.5 ms. The stimulus size for both conditions was the same: 4.1 × 2.7 cm (2.35 × 1.55° at ~100 cm). In the following, the specific characteristics of the stimuli used in each of the paradigms will be described.

In the RFW paradigm, which was based on the proposal by Zhang et al. [4], the stimulus overlaying the white letters was a semi-transparent image of the face of Barack Obama, colored in red with a white surrounding rectangle (Figure 1a). In the NP paradigm, which was inspired by Fernández-Rodríguez et al. [19], each character of the matrix was flashed using a different neutral picture (Figure 1b). In order to ensure that the pictures were neutral, they were obtained from the International Affective Picture System (IAPS) [24]. Pictures with the lowest score for arousal level and that had the aforementioned size (i.e., images that filled all the space without black padding) were chosen.

### 2.4. Procedure

This experiment was carried out in an isolated room. Participants sat a distance of approximately 1 m from the computer screen. Instructions were given in both written and verbal form, and each experiment was performed in a single session through an intrasubject design, during which the same participant used both paradigms (Figure 2). Also, in order to prevent the effect of extraneous variables that could influence the results—such as fatigue or learning—the order in which the paradigms were presented was counterbalanced across the participants.

Immediately after explaining the experiment to the participant, obtaining written consent, and collecting the questionnaire used to gather personal information (such as gender, age, and technological experience) (Test Set 0), a questionnaire with one item related to stress level and the Affect Grid questionnaire were administered (Test Set 1). These elements will be described in detail in Section 2.5. Once these initial questionnaires had been completed, the BCI task could begin.

The evaluation of each paradigm consisted of two parts: (i) a calibration phase, in which the system was adapted to the user; and (ii) a copy-spelling phase, in which the participant wrote four words of four letters each. The subject was free to ask for a pause after each word, in addition to the mandatory pause between the two paradigms. Initially, during the calibration phase, participants were asked to select 12 characters, divided into three words of four letters, in each condition. The number of sequences (i.e., the number of times each row and column were highlighted) was fixed to 10, meaning that each letter was overlaid by a face or a picture (depending on the paradigm) 20 times. The three specific French words used for calibration were: “*FEUX*” (fire), “*CHAT*” (cat), and “*PURE*” (pure). No feedback on the correctness of the spelling was provided to the participants, since the purpose of this phase was to create the classifier. At the end of the calibration phase, a SWLDA was performed to obtain the weights of the P300 classifier. Following this, the copy-spelling phase involved the participant spelling four specific words: “*ABRI*” (shelter), “*LUNE*” (moon), “*YOGA*” (yoga), and “*CHEF*” (boss). The number of sequences used during this phase to select characters was that in which the user obtained the second consecutive best accuracy in the calibration task. In cases where the maximum accuracy was not repeated consecutively or there was only one, the first best sequence was selected. During this phase, each time a participant selected a character, it was displayed at the top of the screen. An incorrect letter could be selected, but the subject was warned that he/she had to continue with the next one.

After completing the copy-spelling phase using each paradigm, the user was asked to complete the SUS (Test Sets 2 and 3). Finally, after finishing all the BCI tasks, the user was again asked to answer the item related to stress level (to measure the change produced by the BCI session) and the Affect Grid, and, for the first time, an ad hoc item concerning fatigue and two items related to the level of pleasantness for each paradigm (Test Set 4).

### 2.5. Evaluation

As described in the introduction, our goal was to evaluate the usability and the performance of two different conditions: famous faces shown in red on a white background (RFW) and neutral pictures (NP). In order to carry out this evaluation, we built our analysis on three dimensions: (i) effectiveness, (ii) efficiency, and (iii) satisfaction.

#### 2.5.1. Effectiveness

Effectiveness denotes the performance with which the user succeeds in the task. Three specific variables were considered in the evaluation of effectiveness: (i) the number of sequences used; (ii) the accuracy; and (iii) the information transfer rate (ITR, bit/min), based on the formula presented by Wolpaw et al. [25]. The ITR is an objective measure used to determine the communication speed of the system, and is expressed as:$$\mathrm{ITR}=\frac{\log_{2}N+P\log_{2}P+\left(1-P\right)\log_{2}\frac{1-P}{N-1}}{T}$$
where *P* denotes the classification accuracy, *N* denotes the number of available characters in the interface (25 in this experiment), and *T* denotes the time needed to perform a selection.

Since the performance of the system depends on the recorded EEG signal, it would be interesting to study the grand average of the ERP waveforms (µV). More specifically, the larger the differences between the amplitudes for the target and non-target stimuli in a given paradigm, the easier it will be for the classifier to discriminate between them. Hence, the two conditions (NP and RFW) were compared using the amplitude difference as a variable (µV, amplitude of the target stimulus minus the amplitude of the non-target stimulus, from 0 to 800 ms). To perform this analysis, a baseline ranging from −200 to 0 ms and low-pass filtered at 30 Hz was used for each electrode.

#### 2.5.2. Efficiency

Efficiency is related to the number of resources needed by the participant to achieve the required task. Two specific variables related to this dimension were collected at specific times during the experiment: (i) stress and (ii) fatigue. The stress level was measured before and after the BCI tasks using the following item (during Test Sets 1 and 4): “Evaluate your current stress level”. The users’ responses were graded on a scale of one to five, with five indicating the highest stress level. Fatigue was measured through two different ad hoc items after performing all the BCI tasks (during Test Set 4). First, users were asked about their level of fatigue: “To what extent did you find this experiment tiring?” The response to this item was on a scale of one to five, with five representing the highest level of fatigue. The second item related to fatigue aimed to find out which paradigm had been the most exhausting for the user: “Which paradigm would you say was the most exhausting?” There were three possible answers to this item: RFW, NP, or equivalent.

#### 2.5.3. Satisfaction

Satisfaction is related to the participants’ attitudes on the system, i.e., their perceived comfort and their well-being. These two notions were explored using three questionnaires: (i) the SUS (Test Sets 2 and 3), (ii) the Affect Grid (during Test Set 4), and (iii) an ad hoc item related to pleasantness (during Test Set 4).

The SUS is commonly used to collect a user’s subjective rating of the usability of a product in a quick and easy format [12]. In the present study, participants were asked to answer the questionnaire after completing the copy-spelling task in both paradigms. This survey contained questions about the convenience, the complexity, and the integration of the functionalities in the interface. The answers permitted us to calculate a global score (of between zero and 100). Bangor et al. [26] explained how to interpret a SUS score of this type. In order to clarify the meaning of this score, seven terms are used: worst imaginable, poor, acceptable, good, excellent, and best imaginable. The minimal score for acceptance is 50.9; however, the probability of acceptance for the system is low when scores fall between this minimal score and about 63. Above 63, the probability is rather higher, and above 71.4, the system’s usability is judged to be good and commonly accepted. The best usability is obtained for a score of above 90.9.

The Affect Grid was developed to quickly assess the current mood of a subject [27]. This questionnaire was administered twice during the experiment, that is, before and after the BCI tasks (Figure 2). Therefore, the purpose of this questionnaire was not so much to establish a comparison between the two paradigms used (RFW versus NP) but between before and after the performance of the BCI tasks in order to assess the effect it could have on the user’s condition. The matrix employed by the Affect Grid should be read along two axes (Figure 3). The abscissa refers to the pleasure level (from unpleasant to pleasant feelings), while the ordinate represents the arousal level (from sleepiness to arousal). Participants were asked to choose the number that best matched their current state. The four corners of the grid correspond to specific feelings: stress, excitement, depression, and relaxation. As explained by Russell et al. [27], two parameters can be extracted from the Affect Grid: the pleasure score (P) and the arousal score (A). The P score corresponds to the column of the number chosen by the participant (between one and nine, counting from the left), while the A score corresponds to the row (between one and nine, counting from the bottom). For example, if a participant estimates his or her mood as corresponding to the number 24, the P and A scores are equal to six and seven, respectively.

Finally, an ad hoc item was used to find out how pleasant the participant found the use of the system under each of the paradigms (“How pleasant was the keyboard to use?”). This item was answered after completing all the BCI tasks and was measured on a scale of one to five points, with five indicating maximum pleasure.

### 2.6. Statistical Analysis

Several statistical analyses were carried out. The aims were to study the potential differences between the two conditions (RFW and NP) and to examine the number of sequences, accuracy, ITR, ERP waveform, the most tiring condition, SUS score, and pleasantness score. There was also one variable (the user’s stress level) that was not compared between conditions, but it was measured at the beginning and end of the experimental session to evaluate whether it was influenced by the overall execution of the experiment.

With the exception of the ERP waveform analysis, the rest of the variables were analyzed using SPSS software (v24) [28]. Before proceeding with each analysis, the assumption of normality was checked before proceeding to a Student’s *t*-test for repeated samples or a Wilcoxon test (except for the most tiring condition variable), depending on whether the criterion was met or not, respectively. The variable that explored which condition had been more tiring was evaluated through a binomial test of the proportion of users who chose one condition over the other. For the ERP waveform, statistical analyses were carried out using EEGLAB (v2022.0) [29], where the options to use parametric statistics and the false discovery rate (FDR) as a correction method for multiple comparisons were selected.

## 3. Results

### 3.1. Effectiveness

#### 3.1.1. BCI Performance

Table 1 shows the number of sequences used, the accuracy, and the ITR for each participant and condition. The average number of sequences used during the copy-spelling phase was 4.83 ± 2.04 for the RFW, and 3.58 ± 1.38 for NP. The average accuracy was 97.42 ± 4.23% for RFW, and 94.83 ± 5.95% for NP. Finally, the average ITR was 22.94 ± 11 bit/min for RFW, and 27.92 ± 10.47 bit/min for NP. No significant differences were found in relation to any of the variables reported: number of sequences (*Z* = 1.621; *p* = 0.105), accuracy (*Z* = 1.089; *p* = 0.276), or ITR (*Z* = 0.941; *p* = 0.347).

#### 3.1.2. ERP Waveform

The ERP waveforms found were within the expected range for a visual ERP-based BCI under the RCP condition. The main component to note was a strong positivity of around 350–600 ms, which was generalized along the scalp surface (Figure 4). Some channels around the parietooccipital region (P4, PO7, PO8, and Oz) also showed positivity (~200 ms), which was immediately followed by a pronounced negativity (~300 ms).

A comparison of the conditions showed no significant differences in any of the registered channels. These results can help us to understand the lack of significant differences in the BCI performance; if the EEG signal does not differ significantly between conditions (RFW and NP), it is difficult for the BCI performance to do so. Nevertheless, a tendency can be observed in the frontal channels (Fz and Cz) of RFW towards higher positivity throughout the temporal interval studied compared to NP, as well as a stronger negative peak in PO7 and PO8 for RFW. The interpretation of the possible components and a comparison with the related literature will be discussed in a later section.

### 3.2. Efficiency

#### 3.2.1. Stress

From the item that measured the participant’s current stress level, we saw that the average stress score before performing any of the BCI tasks was equal to 1.83 ± 1.19 points, whereas this score was equal to 1.58 ± 0.67 points after completing all the BCI tasks. There were no significant differences between the stress levels before and after the experience (*Z* = 1.089; *p* = 0.276).

#### 3.2.2. Fatigue

From the item measuring how tiring the participant found the experiment, we saw that they scored it with an average of 3.42 ± 1 points, on a scale from one to five. From the item asking participants to select the most exhausting condition, the following results were obtained: eight participants chose the RFW condition, three chose the NP condition, and one found the two conditions to be equivalent. A binomial test indicated that there was no statistically significant difference between the proportion of participants who chose one paradigm as the most tiring (eight out of 11 for RFW and three out of 11 for NP; *Z* = 1.206; *p* = 0.227).

### 3.3. Satisfaction

#### 3.3.1. System Usability Scale

The average scores obtained on the SUS after completing the copy-spelling phase for each paradigm were 66.04 ± 12.27 points for RFW and 66.04 ± 12.18 points for NP. Hence, no significant differences were found between the two conditions (*Z* = 0.655; *p* = 0.512).

#### 3.3.2. Affect Grid

Data on each participant’s state of mind, gathered via the Affect Grid before and after all BCI tasks, were analyzed. More precisely, the pleasure and arousal scores were investigated. Figure 5 shows that before the experiment (red points), users’ states were scattered across different regions of the grid; however, after the experiment (blue points), their states were all within the same region, which was related to a state of relaxation (shown with a blue dashed outline in Figure 5).

#### 3.3.3. Pleasantness

After completing all the BCI tasks, participants rated how pleasant the use of RFW and NP had been. On average, RFW scored 2.67 ± 0.89 points, while NP scored 3.92 ± 0.79. In this case, significant differences between conditions were found (*Z* = 2.319; *p* = 0.02). It can therefore be stated that NP was found to be more pleasant to use than RFW.

## 4. Discussion

The aim of this work was to compare two paradigms that had been previously studied, referred to here as RFW and NP. Zhang et al. [4] showed that flashing red faces with a white background led to better results than other studies using famous faces. However, Fernández-Rodríguez et al. [19] examined the valence of pictures and their performance on the P300 speller and proved that the use of pictures, both neutral and with a valence, increased the performance versus the standard flashing of letters from grey to white. Hence, a comparison of both paradigms for the first time might offer valuable insights.

In addition to this comparison, it is important to study the usability of this device, since this has not been conducted in previous studies. The inclusion of usability concerns in the development of a P300 speller could be valuable for research on specific potential users (e.g., ALS patients). In this study, three dimensions of usability were analyzed: (i) effectiveness, (ii) efficiency, and (iii) satisfaction. Our study provides a comparison of RFW and NP based on these dimensions. The following paragraphs discuss the results for each dimension.

### 4.1. Effectiveness

#### 4.1.1. BCI Performance

The present work confirmed the preliminary results reported by Ron-Angevin et al. [21], in that there was no significant difference between the two paradigms in terms of performance. Thus, performance is not a factor that affects the decision on which type of stimulus is more appropriate for the control of a visual P300 speller. In the following, the three variables used to evaluate performance will be discussed: the number of sequences, accuracy, and ITR.

The number of sequences chosen for the copy-spelling phase corresponds to the number of sequences needed to guarantee adequate accuracy, based on the performance of the classifier in the calibration task. Although the results from each paradigm were similar, fewer sequences were necessary to achieve an accuracy of 100% in the NP experiment than in the RFW one. Thus, more sequences were used in the RFW condition than in NP in the copy-spelling phase (4.83 and 3.58 sequences, respectively). The number of sequences could affect other variables related to the writing time, such as the ITR. Hence, as long as the accuracy remains acceptable, a decrease in the number of sequences used should be considered positive, as it will reduce the time required for each selection and will speed up the control of the interface, which could also improve the user experience.

The results for the accuracy also did not indicate that one paradigm was better than the other, with a difference of only 1.5% in favor of RFW versus NP (97.42% and 94.83%, respectively). The question of whether this increase in the case of the RFW condition compensates for the fact that the user needed to use on average 1.25 sequences more than in the NP condition (4.83 and 3.58, respectively) therefore needs to be answered, and we attempt to do this using the following metric.

The amount of information transmitted in a given period of time impacts the quality of communication, so a high ITR is necessary to improve the performance of BCIs. This variable did not show significant differences between conditions. However, it should be noted that the ITR of NP was 21.71% (4.98 bit/min) higher than that of RFW. Hence, although it was not significant, NP had a somewhat higher ITR than RFW (27.92 and 22.94 bit/min, respectively).

Our results for the effectiveness are compared with those of Zhang et al. [4] and Fernández-Rodríguez et al. [19] in Table 2. Since we used the stimuli proposed in both these studies, it would be interesting to compare the accuracy and ITR for the three schemes. The results reported by Zhang et al. [4] and our results were almost the same for the accuracy (96.94% and 97.42%, respectively); we therefore note that their results were replicated and that a RFW paradigm is indeed able to provide high levels of accuracy. Since we found that there were no significant differences between the RFW and NP paradigms, it can also be concluded that the NP paradigm had a high classification accuracy. However, the results obtained by Fernández-Rodríguez et al. [19] and the present study for the NP paradigm were quite different, in terms of both accuracy and ITR (Table 2). This could be explained by the differences between the studies, since the conditions differed in certain respects such as the number of possible selectable elements and the size of the interface, parameters that could influence the performance of a P300 speller [30,31].

#### 4.1.2. ERP Waveform

As mentioned in the Results section, the ERP waveforms found in this study were in line with those in the literature. However, it should be noted that researchers focusing on BCIs aim to optimize system performance and maximize the user experience, and these devices are not intended to provide theoretical answers about ERP components; neither the presentation paradigm nor the protocol were designed for this purpose. This means that the interpretations provided below must be considered carefully. The positivity found at around 350–600 ms, which was generalized across the entire scalp surface, was probably a P300 signal, the component that gives its name to this type of BCI system (i.e., a P300 speller). This positivity has been also reported at similar temporal intervals in several other studies (e.g., by Kellicut-Jones and Sellers [15] and M. Li et al. [32]). The negativity found for the occipital areas (PO7, PO8, and Oz) at around 280 ms could be an N170 signal (e.g., Kaufmann et al. [16], Q. Li et al. [17], and Lu et al. [33]). The N170 has been associated with image processing, and particularly with face processing (e.g., Eimer [34] and Tian et al. [35]). This would make sense considering that this component was larger for RFW than for NP (although this was not a significant effect).

### 4.2. Efficiency

#### 4.2.1. Stress

Regarding the items related to stress, it should be noted that most of subjects (nine out of 12) had the same level of stress before and after the BCI tasks. Of the three participants whose stress scores varied, one (participant S2) became more stressed after the experiment (from one to two points) and the two others (participants S3 and S11) were more stressed before the experiment (with stress levels of five and three points, respectively) and became more relaxed afterwards (three and one points, respectively). However, the least stressed subjects did not achieve better results in terms of performance. Despite some variations, we conclude that this experiment had no effect on stress, as demonstrated by the statistical analysis (*p* = 0.276). The average stress scores before and after the experiment were 1.83 and 1.58 points, respectively.

#### 4.2.2. Fatigue

The average level of fatigue was higher than three points (3.42 ± 1) on a scale from one to five, which could indicate that participants found the experiment tiring. Participants S1, S3, S10, S11, and S12 felt very tired (four or five points out of five). These results are important, since the use of BCI systems may be challenging and require a lot of effort, especially for their target population (i.e., people with severely impaired motor skills). Thus, reducing the impact on fatigue must be a priority in future studies. If one of the two paradigms had to be chosen, the NP paradigm should be considered, even though the differences were not significant, since 66.67% of the participants found it less exhausting than the RFW paradigm.

### 4.3. Satisfaction

#### 4.3.1. System Usability Scale

According to Bangor et al. [26], the scores obtained from the SUS allow us to establish a series of thresholds to classify the usability of the system (“worst imaginable”, “poor”, “acceptable”, “good”, “excellent”, and “best imaginable”). In the present study, the two paradigms showed similar scores (66.04 points for both conditions). In particular, since both of these scores were above 50.9 points, each of the two conditions can be labeled as acceptable (the threshold for labeling the system as “good” is 71.4). There is still a wide margin for improvement to increase the usability of these systems, and, as stated above, future studies should make efforts in this direction, rather than only focusing on the dimensions related to performance.

#### 4.3.2. Affect Grid

As shown in Figure 5, after the experiment, the results for all subjects were gathered in the same region. This region corresponds to pleasant feelings (high pleasure), sleepiness (low arousal), or even relaxation (high pleasure with low arousal) for some participants, meaning that in general, participants tended towards a state of well-being at the end of the experiment. Although most participants were more tired after the experiment, we also noted that some of them felt better (moving from a depressed state to a lightly relaxed one). For some, there was a reduction in their well-being, but they were still in a neutral/relaxed state after the experiment. However, the state of the users generally improved (from stress to relaxation). It is possible that this improvement towards a relaxed state was due to the initial excitement of the participant when entering an experimental scenario, and that as he/she became habituated to the context and acquired confidence in using the system, this state gradually improved. In short, it seems that using the system at least did not worsen the user’s initial state.

#### 4.3.3. Pleasantness

The NP paradigm turned out to be a more pleasant paradigm than RFW (3.92 and 2.67 points, respectively), and significant differences were found between the two. This result is particularly interesting, since a P300 speller is intended to establish a communication channel for people with severe motor impairments who cannot use speech or other alternative methods requiring muscle mobility [1]. Thus, these systems must be designed for daily use, meaning that the designer should not only consider performance measures, but also others such as the user’s pleasure.

The study by Zhang et al. [4] presented a type of visual stimulus—those employed in our RFW paradigm—that offered promising accuracy performance on a P300-BCI visual. Nevertheless, the RFW paradigm may be less usable than the NP paradigm. Indeed, as mentioned above, in the NP paradigm, a pictogram or command can be used to facilitate communication through the system. Our findings for satisfaction support the view that the NP paradigm could have more advantages than the RFW paradigm.

## 5. Conclusions

The aim of the present work is to assess the use of two different types of stimuli in a visual ERP-based BCI under RCP for communication purposes: red faces surrounded by a white rectangle, which was reported to yield the best performance in previous work [4], and neutral pictures, which have also shown promising performance in a prior study [19]. In general terms, it seems that both types of stimuli tend to receive a positive evaluation for objective and subjective measurements; this means that both could be useful stimuli when implemented in a P300 speller to control such equipment as a home automation system or any other application that can improve the quality of life of patients with severe motor impairments. However, NP was found to be significantly more pleasant to use than RFW. Hence, in the absence of significant differences in measures related to performance, it seems appropriate that the choice of one paradigm over another should depend on subjective parameters, such as the pleasantness felt while using the system. After all, systems such as those for home automation are intended to be controlled by patients on a daily basis, so it is important to improve the overall user experience (since a system that is not pleasant to use may not be attractive for daily use, even if it gives adequate performance). Likewise, other characteristics such as the specific purpose of the system should be considered; for example, the NP paradigm allows for the use of different images such as pictograms as stimuli, and when these are related to the desired command (e.g., an image of a television if the user wants to use it) the system may be more intuitive and comfortable than one that uses unrelated images.

In short, the present work demonstrates the usefulness of the NP paradigm in terms of the higher pleasure of the user during system control compared to RFW, with no differences in performance. It also demonstrates the importance of evaluating not only performance but also the factors affecting the complete experience through a usability analysis, which is especially important to offer this technology to potential users in their daily lives.

## Figures and Tables

**Figure 1 sensors-23-01304-f001:**
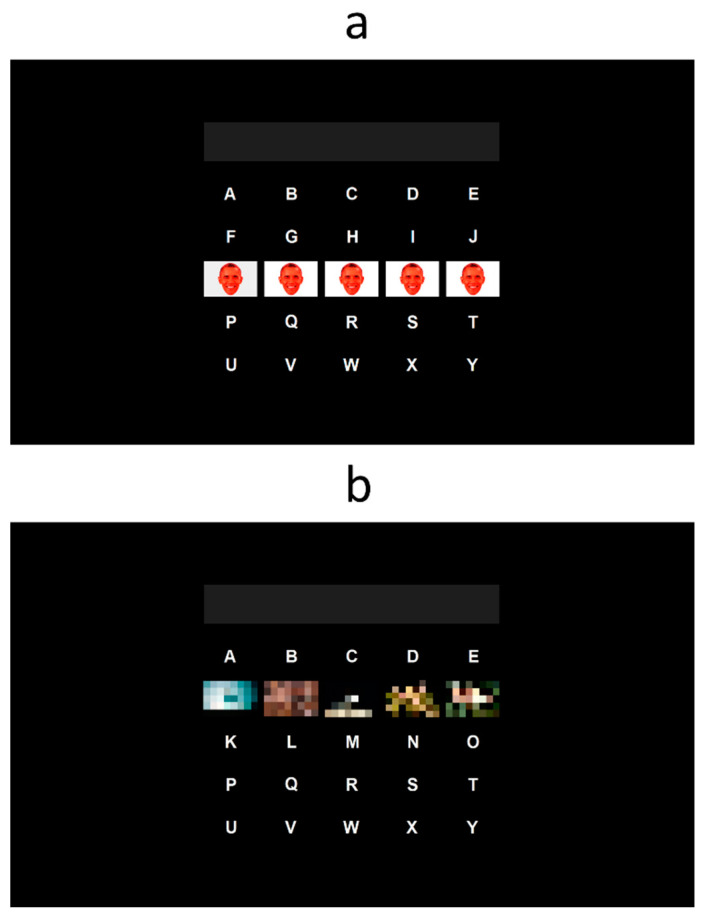
Two spelling paradigms were tested in the experiment, which varied in terms of the flashing stimulus used: (**a**) famous faces shown in red with a white rectangle (RFW), based on [4]; and (**b**) neutral pictures (NP), based on [19]. For copyright reasons, the neutral pictures have been pixelated here.

**Figure 2 sensors-23-01304-f002:**
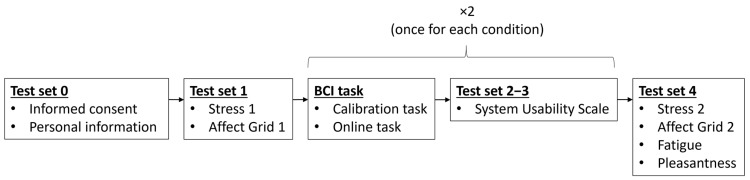
Sequence of questionnaires and tasks completed by each participant during the experimental session.

**Figure 3 sensors-23-01304-f003:**
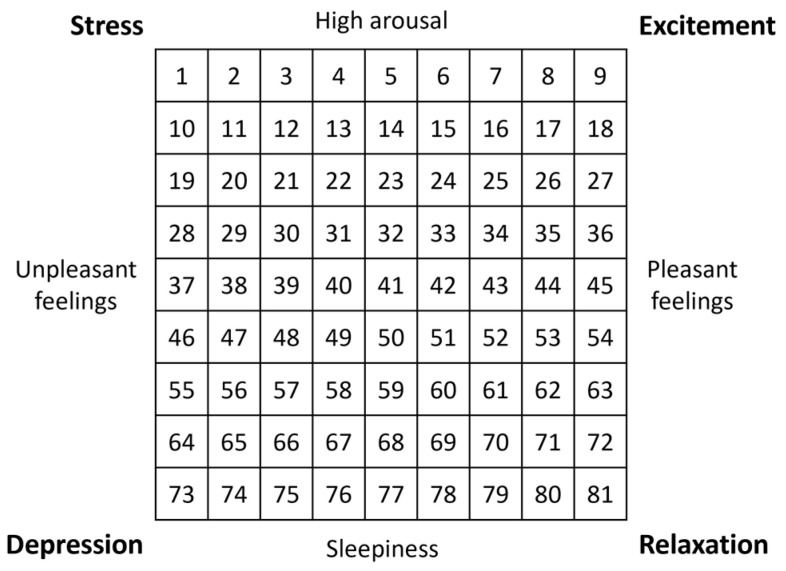
The matrix employed in the Affect Grid questionnaire to indicate the participant’s affective state using two dimensions (pleasantness and arousal).

**Figure 4 sensors-23-01304-f004:**
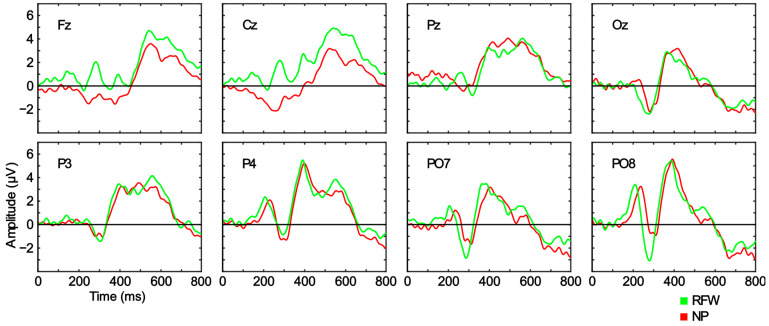
Grand average event-related potential (ERP) waveforms for the amplitude differences (μV) between the target and non-target stimuli for each condition (red faces surrounded by a white rectangle (RFW) and neutral pictures (NP)) and channel (Fz, Cz, Pz, Oz, P3, P4, PO7, and PO8), represented over the time interval from 0 to 800 ms. No significant differences were found between the target and non-target stimuli for any of the channels.

**Figure 5 sensors-23-01304-f005:**
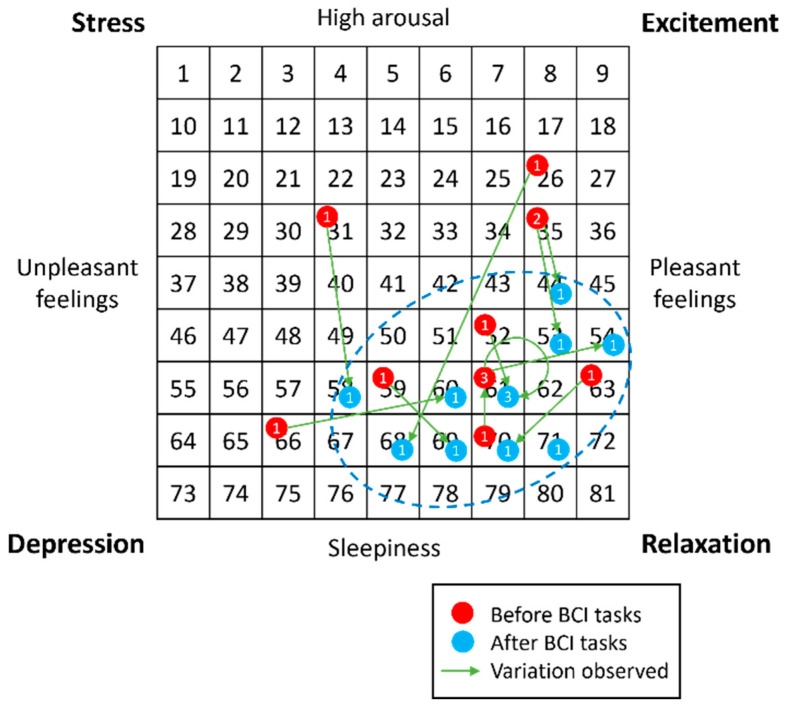
Results for each participant on the Affect Grid questionnaire, before and after the brain–computer interface (BCI) tasks. The number inside each circle refers to the number of participants who obtained that score.

**Table 1 sensors-23-01304-t001:** Results for each participant, and average (±standard deviation), in the copy-spelling phase in terms of number of sequences used, accuracy (%), and ITR (information transfer rate, bit/min) for each condition (red faces surrounded by a white rectangle (RFW) and neutral pictures (NP)).

User	Number of Sequences		Accuracy (%)		ITR (bit/min)	
	RFW	NP	RFW	NP	RFW	NP
S1	3	7	100	100	33.02	14.15
S2	6	2	87	100	12.41	49.53
S3	2	3	100	100	49.53	33.02
S4	5	5	100	81	19.81	13.11
S5	10	4	100	94	9.91	21.55
S6	4	3	94	94	21.55	28.74
S7	5	4	100	100	19.81	24.77
S8	5	3	94	94	17.24	28.74
S9	5	3	100	94	19.81	28.74
S10	3	4	100	100	33.02	24.77
S11	6	2	94	94	14.37	43.11
S12	4	3	100	87	24.77	24.82
Average	4.83 ± 2.04	3.58 ± 1.38	97.42 ± 4.23	94.83 ± 5.95	22.94 ± 11	27.92 ± 10.47

**Table 2 sensors-23-01304-t002:** Comparison of average performance for the conditions considered here and the results of other studies.

Work	Accuracy (%)	ITR (bit/min)
Present study: red faces with a white rectangle (RFW)	97.42	22.94
Present study: neutral pictures (NP)	94.83	27.92
Zhang et al. [4]	96.94	-
Fernández-Rodríguez et al. [19]	99.17	42.6

## Data Availability

The data presented in this study are available on request from the corresponding author.

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
