# Peer review of "Comparison of Two Paradigms Based on Stimulation with Images in a Spelling Brain–Computer Interface"

_sensors, 2023, doi:10.3390/s23031304_

Round 1

Reviewer 1 Report

1. Many similar studies have evaluated the use of color and types of stimulant images. Comparing the two stimulants is the less significant novelty and expediency unless the reason is explained by usability, effectiveness, and system performance efficiency (according to Table 1 and Figure 4, such accuracy).

2. How does this research ensure that the tests are carried out solely due to using BCI? Moreover, how to validate it. Given that it is not easy to answer various questionnaires. How is the selection of research subjects?

3. What does each column in Table 1 mean

4. Figure 5 should better explain the comparison of the two stimulants

Reviewer 2 Report

Thank you for giving me the opportunity to review the manuscript "Comparison of two paradigms based on stimulation with images in a spelling brain-computer interface: by Ricardo Ron-Angevin, Álvaro Fernández-Rodríguez, Clara Dupont, Jeanne Maigrot, Juliette Meunier, Hugo Tavard, Véronique Lespinet-Najib, Jean-Marc André.

The paper's first two authors are the UMA-BCI Speller's designers for BCI2000. They joined their efforts with a French team from L’Ecole Nationale Supérieure de Cognitique (ENSC) to collect data from human subjects using their speller and BCI2000. 

They hypothesized that there is no systematic difference between the performance of the speller with a famous face colored in red with a surrounding square of a specific color (Zhang et al. study) versus neutral pictures (Fernández-Rodríguez et al. study).

This study is a more comprehensive assessment. The team already conducted and published preliminary results to the Seventh International Conference on Neuroscience and Cognitive 588 Brain Information; Venice, Italy, 2022. They also included in this study measures of usability.

The study is significant as it clarifies a long-lasting question regarding the effectiveness of different protocols for the P300 speller. They found that there were no significant improvements in spelling accuracy regardless of the protocol.

I support the publication of the manuscript as is.

Author Response

Thank you very much for taking the time to review the document. We are pleased to read that you found interesting the idea of the paper on comparing two paradigms with high performance using a comprehensive assessment.